# Crowd medical services in the English Football League: remodelling the team for the 21st century using a realist approach

Alison Leary,[1] Anthony Kemp,[2,3] Peter Greenwood,[4] Nick Hart,[5] James Agnew,[6] John Barrett,[6] Geoffrey Punshon[1]

[1]School of Health and Social Care, London South Bank University, London, UK
[2]School of Healthcare Practice, University of Bedfordshire, Luton, UK
[3]Britsih Association for Immediate Care, Ipswich, UK
[4]St. John Ambulance, London, UK
[5]Association of Millwall Supporters, London, UK
[6]London Ambulance Service NHS Trust, London, UK

**Correspondence to**
Dr Geoffrey Punshon;
punshongeoff@yahoo.co.uk

## ABSTRACT

**Objectives** To evaluate the new model of providing care based on demand. This included reconfiguration of the workforce to manage workforce supply challenges and meet demand without compromising the quality of care.
**Design** Currently the Sports Ground Safety Authority recommends the provision of crowd medical cover at English Football League stadia. The guidance on provision of services has focused on extreme circumstances such as the Hillsborough disaster in 1989, while the majority of demand on present-day services is from patients with minor injuries, exacerbations of injuries and pre-existing conditions. A new model of care was introduced in the 2009/2010 season to better meet demand. A realist approach was taken. Data on each episode of care were collected over 14 consecutive football league seasons at Millwall FC divided into two periods, preimplementation of changes and postimplementation of changes. Data on workforce retention and volunteer satisfaction were also collected.
**Setting** The data were obtained from one professional football league team (Millwall FC) located in London, UK.
**Primary and secondary outcomes** The primary outcome was to examine the demand for crowd medical services. The secondary outcome was to remodel the service to meet these demands.
**Results** In total, 981 episodes of care were recorded over the evaluation period of 14 years. The groups presenting, demographic and type of presentation did not change over the evaluation. First aiders were involved in 87.7% of episodes of care, nurses in 44.4% and doctors 17.8%. There was a downward trend in referrals to hospital. Workforce feedback was positive.
**Conclusions** The new workforce model has met increased service demands while reducing the number of referrals to acute care. It involves the first aid workforce in more complex care and key decision-making and provides a flexible registered healthcare professional team to optimise the skill mix of the team.

## INTRODUCTION

Crowd medical services in the English Football League were formalised by the recommendations of Lord Justice Taylor[1] following an incident at Hillsborough Stadium, in which 96 spectators died in 1989, and the subsequent findings of the enquiries which have been ongoing. In 2012, the inquest into the deaths at Hillsborough was reopened.

After the recommendations of the Taylor Report[1] and previous legislation,[2–4] the current Guide to Safety at Sports Grounds[5] was published. This guidance is often referred to as the *Green Guide*. It is not statute but incorporates many of the acts that relate to crowd safety within sports stadia in England.

The workforce provision in the guide is for one first aider per thousand spectators, and where a crowd is expected to exceed 2000 a crowd doctor who is trained in immediate care should be provided. A first aider is defined as an individual holding a first aid certificate, in England they are not usually a healthcare professional, but a lay person with skills such as immediate treatment of bleeding, minor injuries and basic life support. Ambulance provision is included in the guidelines with crowds between 5000 and up to 25 000 requiring ambulance with paramedic crew. Statutory ambulance provision increases with crowds between 25 000 and 40 000 and again >40 000.[6] Despite a thorough search

of the literature, no evidence for these ratios or educational standards has been found and it appears they were arrived at by consensus at the time.

### Strengths and limitations of this study

► The study monitored crowd medical services over a 14-year period.
► The study demonstrates that the demand for crowd medical services is more varied than the requirements of the 'Green Guide' and that remodelling of the service was necessary.
► The workforce responded positively to the changes made to the service.
► The study was limited by incomplete data collection in the early years of the study and as it was carried out at a single football league team.

The Hillsborough disaster influenced much of the framework for the guidance of provision of crowd medical services in league football over the next 25 years and for good reason. Lack of triage and immediate scene management by the ambulance service caused or contributed to the loss of lives by failing to recognise or actuate a major incident.[7] Subsequently, the guidance on provision of services has focused on extreme circumstances such as mass casualty situations and physical environments such as all standing crowds, which largely no longer exist in the football league. This has meant that guidance for the design of medical service provision is led by a 'black swan', a rare event with extreme impact and retrospective predictability[8] while the majority of demand is given a lesser priority and resource but is a far more likely to have a greater impact on the service.[9–11]

The demand on services in present-day spectator care is rarely from major mass casualty situations. Commonly, demand is from patients with minor injuries, exacerbations of illness, pre-existing conditions and occasionally emergent patients.[9–12] Designing a service model that can accommodate both immediate disaster management but also the higher volume of minor injuries, medical emergencies and primary care work presents a different challenge to that designed into national guidance such as the *Green Guide*'. In recent years, austerity measures in England have also placed resource constraints on healthcare service providers such as the statutory ambulance service and the acute sector.[13] Managing demand at source has become a fundamental necessity.

Millwall's ground, The Den, was built in 1993 as part of the post-Taylor initiative[14] with a capacity of 20 146. The club is currently located between two inner London boroughs, Lewisham and Southwark. Residents are more likely to die an early death through cancer, heart disease or smoking-related illnesses, and in Lewisham have a life expectancy 6.8 years lower than the England average.[15]

Planning the medical services for mass gatherings is difficult. The number of variables is complex and their interactions dynamic.[16 17] In order to manage demand in an effective and sustainable way, the service at Millwall was reviewed based on the previous set of demand and outcome data.[9] Concurrently supply of workforce was examined, and a number of local issues were revealed.

### Problem description and rationale for change
A retrospective study had already been carried out,[9] and this identified that the majority of local demand is from non-mass casualty situations such as exacerbations of chronic disease, minor injuries and much less commonly, emergent patients. The statutory requirement to have ambulance vehicles on site was becoming more challenging due to issues with availability of crews and on occasions it was not possible to meet this requirement. This also applied to recruiting individuals to fill the

'crowd doctor' role as changes in training in England has impacted on the availability of supply. Doctors with appropriate training in immediate care and also able to commit to regular rotas proved difficult. Some authors recommend this level of care becomes a prehospital speciality[18] which would present even more challenges in terms of supply. Times of high demand and the working environment demonstrated that skills and attributes beyond technical competency were required and that this particularly applied to the 'crowd doctor' role. The only requirement to become a 'crowd doctor' was to have General Medical Council registration and completion of a 2.5-day Football Association Faculty of Pre-hospital Care Crowd Doctor course, but extreme situations at Millwall demonstrated that this was not sufficient preparation for the role.

Staff turnover was high and inconsistent due to the employment model of ad hoc sessional work. This leads to little team cohesiveness as members worked together infrequently and were not familiar with local working conditions.

Since the publication of the Taylor Report almost 30 years ago, a number of other professional groups such as nurses, paramedics and physiotherapists practice at a much more complex level incorporating advanced practice skills[19] which were not used in the service to any great degree. A re-examination of local data indicated that the default of the first-line treatment by first aiders had a low referral threshold to acute emergency care when it might not be clinically necessary if a healthcare professional was available for advice or review. These findings echo those of Kemp in other event medical services.[12] Despite the presence of the statutory ambulance service at games, the *Green Guide* stipulated this was for major incident use only. In addition to statutory ambulance provision, a vehicle was also provided by a voluntary service agency, St John Ambulance (SJA). SJA increasingly had difficulty providing ambulance cover for games due to a limited supply of volunteers qualified to do this work.

These challenges required a pragmatic response in terms of service redesign and workforce supply in order to manage risk and use the limited resources more effectively. There is evidence to show that high-performing teams and high-reliability organisations[20] have certain attributes, and alongside remodelling the service and workforce supply an approach to examining team makeup was also undertaken.

Examining the teams' effectiveness using the work of Michael West[21 22] revealed a level of high-task reflexivity. The teams were technically focused but the unstable and temporary working patterns characteristic of these services meant a lower-level social reflexivity. This meant a focus on the technical task and less awareness of the situation or of team member's needs. The professionally qualified members of the team such as doctors and paramedics were engaged on a per game basis, meaning very high levels of turnover and unfamiliarity with working practices and the environment. This has on occasions caused serious issues, for example, a major incident in

2002[23] in which the clinical leadership, who were transient, were unsure of their roles despite being qualified to the standard of the then Football Licencing Authority *Green Guide*.[5]

The transient workforce also meant limited professional support was given to the more stable volunteer workforce of first aiders, for example, supporting them to use evidence-based practice in areas such as wound care, medicines management, assessment of traumatic injury and infection control.

### Specific aims

► to reconfigure the workforce to manage workforce supply challenges and yet meet demand without compromising the quality of care;
► to provide capacity and increase activity in other areas such as health promotion.

## METHODS

A medical advisory group of stakeholders (including supporters) was convened to consider all challenges and possible solutions. This group then reported to the overarching Safety Advisory Group which is led by the local authority who grants the safety licence without which the stadium cannot open to the public.

A quality improvement approach was taken. A defining characteristic of quality improvement projects is that they are established primarily as improvement activities rather than research. The principal aim of a quality improvement project is to secure positive change in an identified service.[24] The format taken was an iterative one using the Plan, Do, Study, Act cycle over seven seasons.

After the assessment of the challenges, the response was to undertake a planned implementation of several interventions providing they were approved by the Safety Advisory Group.

The overall approach to change was adoption of Safety II principles[25] focusing on what works well within the stakeholder group. The evaluation used a realist evaluation framework[26 27] using primarily longitudinal observational data to look at context and outcomes but within the mechanism of social change. Realist evaluation is helpful in this kind of project as it is inductive rather than reductive and method agnostic allowing for the narrative synthesis of the different types of data generated and suited to a local study within a specific context.

### Interventions

After historical data were examined to assess demand and the assessment of team reflexivity had been undertaken, several interventions were implemented and are shown in figure 1A.

The workforce changes included formalising a medical coordinator role (a consistent leadership position accountable to the safety officer), discontinuation of the 'crowd doctor' role and subsequent employment of a multidisciplinary team of physicians and nurse practitioners with prehospital qualifications and the skills and attributes to meet the demand.

The medical coordinator is the accountable officer reporting directly to the safety officer. The responsibilities include ensuring staffing and equipment requirements are met, overseeing the medical plan, liaising with the stakeholders, clinical audit and leadership on match days.

A non-hierarchical structure using the formation of a self-organised team that decides its own workflow was agreed alongside 'red rules' to maintain safety.[28] Red rules are safety rules which must never be broken and are commonly used in other safety-critical workforces. This resulted in devolved front-line decision-making that could be supported with further technical skill or clinical acumen if required. A fundamental aspect of these changes was inclusion of the voluntary first aid workforce in strategic decision-making which they had not been involved in before despite being the main provider of care.

Support from a more consistent healthcare professional workforce enabled evidence-based practice to be introduced across the service including within the first aid volunteer workforce (eg, wound care and infection control) as they provide most of the care. There was also an added benefit of senior clinical advice being readily available if required.

The ambulance vehicles which were proving hard to resource and were of very restricted use were discontinued. Attendance of a London Ambulance Service officer at each match who has a primary role to manage a major incident was continued and is fulfilled by a small number of local officers to increase team cohesion.

The *Green Guide* minimum staffing shown in figure 1B was replaced by two registered healthcare professionals and a medical coordinator (also a healthcare professional) per game, an ambulance service officer and one first aider per thousand spectators as illustrated in figure 1C. According to previous data, a crowd of >12 000 is more likely to require an emergency response,[9] thus for games where the expected spectators exceed 12 000 additional resources are present, for example, extra paramedics.

The stakeholder group overseeing change consisted of healthcare professionals, first aid volunteers, London Ambulance Service, representatives from the supporter's club and other stakeholders such as club staff. All decisions/changes were reviewed by the statutory local Safety Advisory Group which is the group given responsibility for safety including issuing of the safety licence.

The service applied for and was granted membership of the British Association of Immediate Care (BASICS) which provided a framework for standards of education and guidance on evidence-based care as well as equipment usage.

### Measures and analysis

Observational longitudinal data were used. The period assessed consisted of 14 consecutive football league

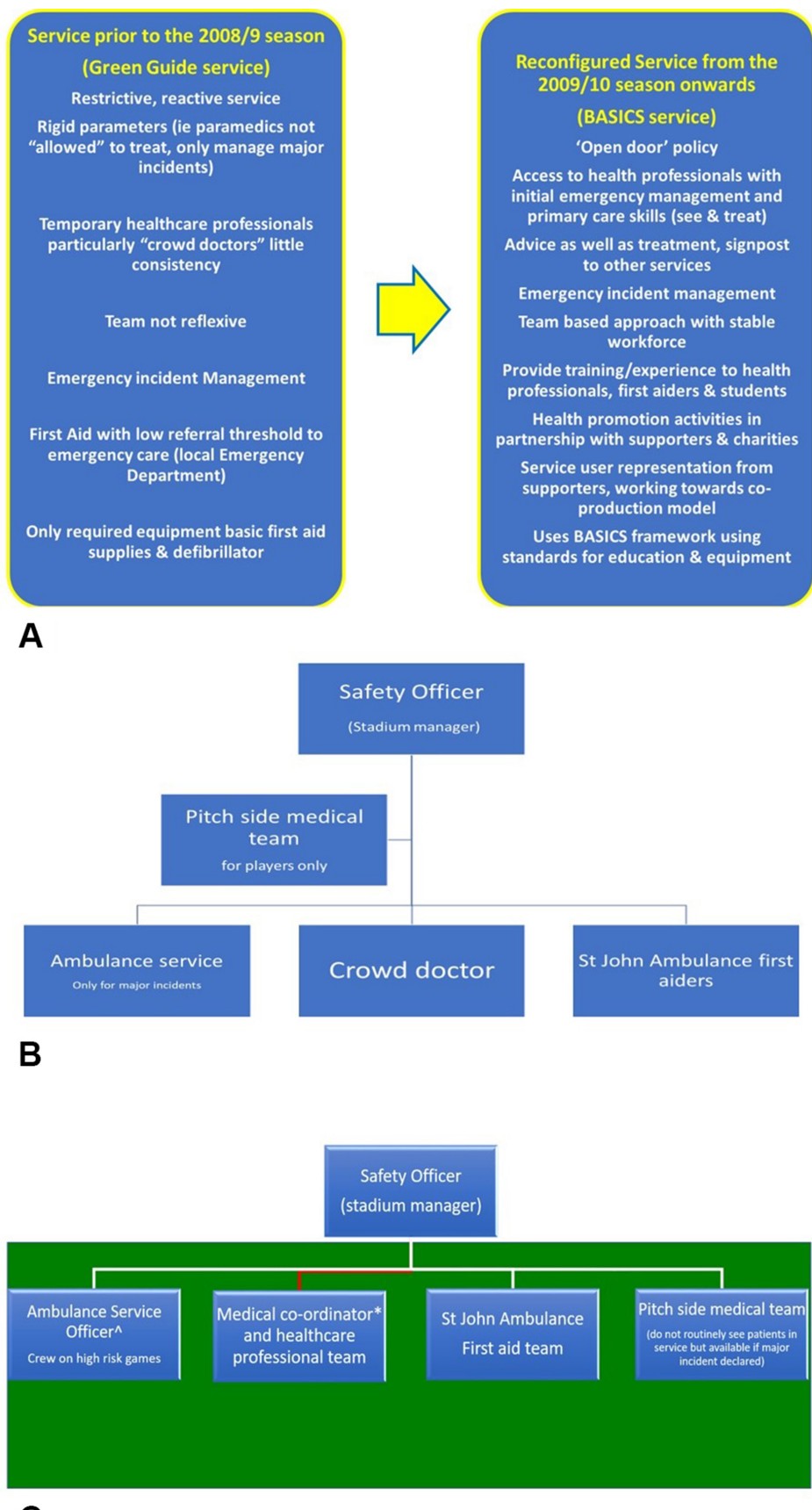

**Figure 1** (A) The medical service prior to the 2008/2009 season and post the 2009/2010 season showing the changes implemented. (B) Organisation of the *Green Guide* service. (C) Organisation of the British Association of Immediate Care (BASICS) service. *Accountable officer for service; ^accountable officer in event of major incident.

seasons at Millwall FC (preimplementation of changes, seasons 2002/2003 to 2008/2009 where care was delivered according to the *Green Guide* guidance and postimplementation seasons 2009/2010 to 2015/2016 where care was delivered within the new framework). A prospective observational study was carried out which employed consecutive sampling to collect data. In the 2009–2010 season, the new workforce model was introduced and so there is a focus in the presentation of data of two phases using descriptive statistics.

The primary outcome measures were usage, skill mix and clinical outcomes. In order to collect the data, an instrument was designed which was used to record each episode of care, consultation or advice given in the regular football league season (ie, not including playoff or exhibition games). This instrument has been previously described[9] and, briefly, collected data on the following: age, sex, postcode (or area of residence), reason for attendance, category (staff or spectator), presenting signs and symptoms, diagnosis, treatment given and outcome. In addition, the skill mix involved in each episode was also recorded. All users of the service were eligible for inclusion in the evaluation.

The data were recorded by the healthcare provider and collated at the end of each match by the medical coordinator.

Data on workforce retention and volunteer workforce satisfaction were also examined.

All data were analysed for activity using an Excel worksheet. As the study design is one of activity/needs analysis, statistical manipulation offers limited benefit and so is limited to descriptive statistics.

## RESULTS
A total of 981 episodes of care were recorded over the duration of the evaluation (392 for the period 2002/2003 to 2008/2009 and 589 for the period 2009/2010 to 2015/2016). Overall the usage of the service increased in

the phase post implementation. This was 0.174–0.33 per 1000 attendances in the preimplementation phase and 0.284–0.452 in the postimplementation phase. This can be seen in figure 2.

### Consultation type
Over the entire time period, 977 episodes of care were characterised as either pre-existing or new conditions. 55.5% of the episodes of care were classified as pre-existing with the remaining 44.5% being new conditions. Across the 14 seasons, the proportion of presentations for pre-existing conditions ranged from 32% to 72%. This is shown in figure 3A.

### Age of users
The age of the user (either the actual age or a general category of adult or child (16 and under)) presenting was recorded in 753 episodes of care. In the remaining 228 presentations, age was either not recorded or the user did not wish to give an age. Of the 711 users who gave an exact age, the youngest was aged 1 and the oldest 92 with a mean age 32.

### Gender of users
The gender of the service users was recorded for 813 episodes of care. On 168 occasions, the gender was not recorded. The percentage of users by gender was 35% female and 64% male.

### User profile
Users were categorised as 'public' (ie, supporters), 'MFC steward', 'MFC staff' (including catering staff, office staff), 'police' or 'player'. The user profile was recorded for 954 episodes of care with 27 episodes where the user profile was either not recorded or unknown. Figure 3B shows the % users for each category in total and for the 2002–2008 and 2009–2015 seasons.

### Reason for presentation
In total, 981 episodes of care were categorised as either 'medical' or 'trauma'. Over the entire study, 57.2% of

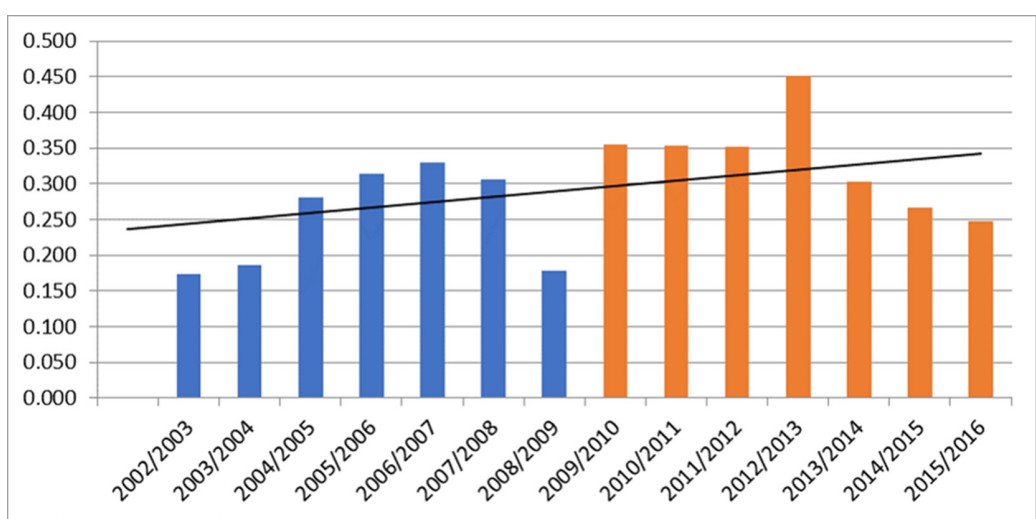

**Figure 2** Total episodes of care delivered per 1000 attendances in the preimplementation and postimplementation phase.

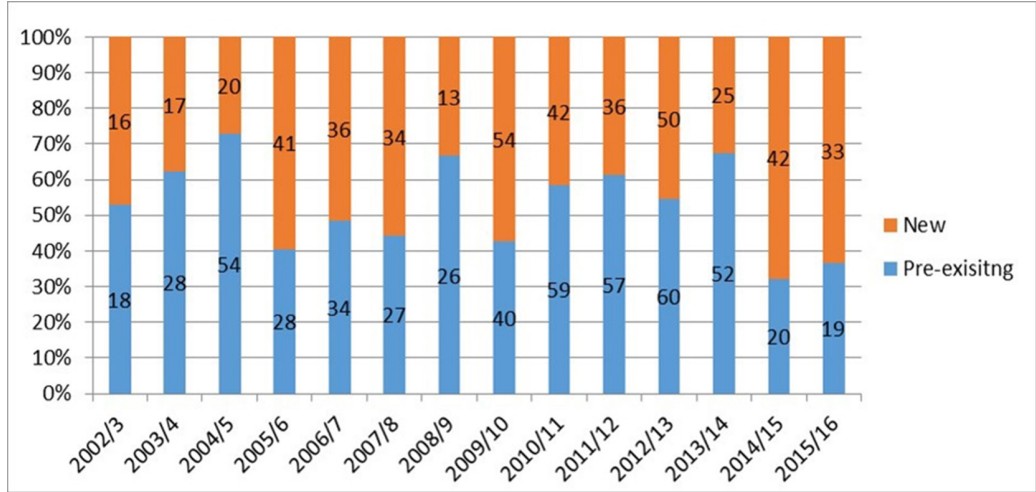

**A**

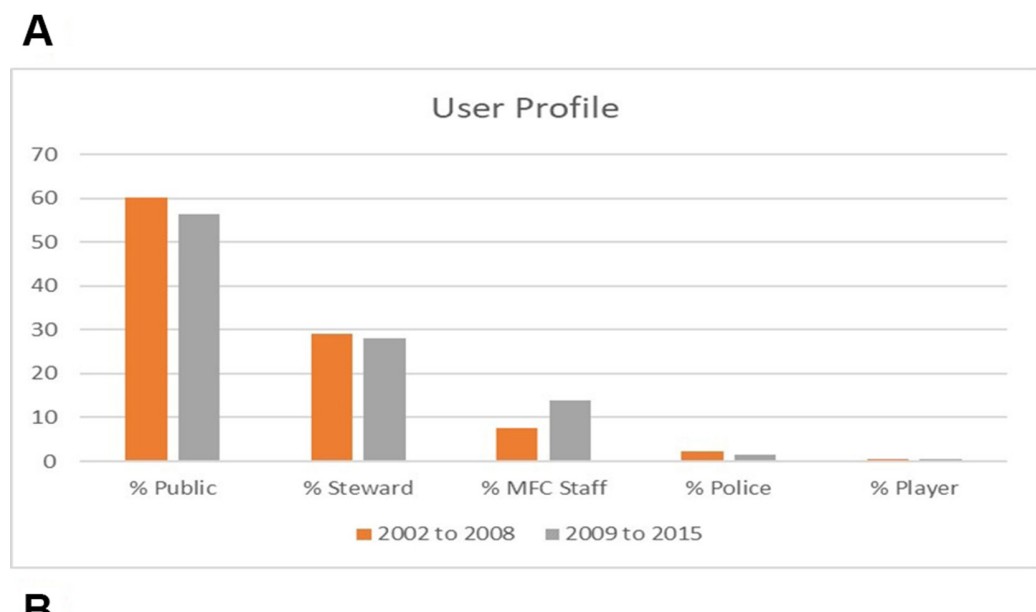

**B**

**Figure 3** (A) Presentations of new and pre-existing conditions—every season saw a significant proportion of presentations from patients with pre-existing conditions. (B) % profile of service users for the total study, 2002–2008 and 2009–2015.

the episodes were categorised as medical with 42.3% categorised as trauma. Between 2002 and 2008, 62.6% of episodes of care were characterised as 'medical', with the remaining 37.4% being 'trauma'. From 2009 to 2015, 52.7% of episodes of care were categorised as 'medical' with 47.3% being 'trauma'.

### Skill mix utilisation

The skill mix of the medical team was logged for each episode of care. For the 2002/2003 season, the categories 'first aider' and 'health professional' (ie, nurse, paramedic or doctor) were used. Following this season, more detailed categories were used with 'first aider', 'nurse', 'doctor', 'paramedic' and 'carer' being used. If more than one group was involved in care, this was recorded as such (eg, 'first aider plus nurse'). Results are presented as a percentage of the total number of episodes for each period.

Over the entire study, 855 episodes of care were recorded (267 for the period 2002–2008 and 588 for the period 2009–2015). On 126 occasions, the skill mix was not recorded.

First aiders alone accounted for 45% of the total recorded episodes of care (45.3% for the period 2002–2008 and 44.9% for the period 2009–2015). First aider plus nurse accounted for 21.4% of the total episodes of care (19.9% for the period 2002–2008 and 26.9% for the period 2009–2015). Nine per cent of episodes of care were provided by a nurse alone (10.1% for the period 2002–2008 and 8.5% for the period 2009–2015). First aider, nurse and doctor dealt with 7.8% of the episodes of care (9.4% for the period 2002–2008 and 7.1% for the period 2009–2015). First aider and doctor accounted for 6.4% of episodes (3% for the period 2002–2008 and 8% for the period 2009–2015). All other combinations accounted for <2% each of the episodes of care. Figure 4

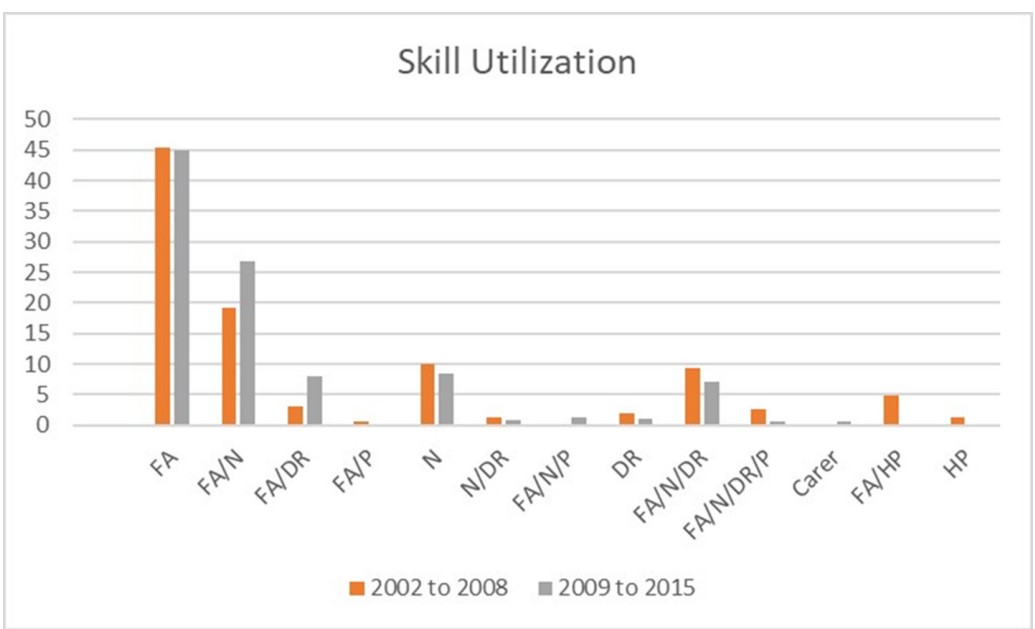

**Figure 4** Skill mix utilisation recorded for each episode of care for the total period (855 episodes), 2002–2008 seasons (267 episodes) and 2009–2015 seasons (588 episodes). Figures are the total % over the period for each category. DR, crowd doctor; FA, first aider; HP, health professional; N, nurse; P, paramedic.

shows the skill utilisation for the total study, the 2002–2008 seasons and the 2009–2015 seasons.

Looking at overall involvement in care, first aiders were involved either alone or with other health professionals in 87.7% of the episodes of care (85.2% for the period 2002–2008 and 89% for the period 2009–2015). Nurses were involved in 44.4% of episodes (44.2% for the period 2002–2008 and 45.4% for the period 2009–2015) while doctors were involved in 17.8% (18.3% for the period 2002–2008 and 17.6% for the period 2009–2015).

### Outcome of episode
The outcome of each episode of care was divided into a number of categories:
► Stay: Patient stayed in the ground (ie, returned to the game).
► Stay+30: Patient stayed in the ground after being in the first aid room for 30 min or longer.
► Stay + general practitioner (GP): Patient stayed in the ground but was advised to visit a GP later.
► Home+GP: Patient went home immediately after the consultation and was advised to visit a GP later if appropriate.
► Minor Injuries Unit (MIU): Patient was sent to a local MIU (an urgent care/walk in facility).
► Hospital: Patient was sent to a local A&E department via an ambulance.
► Custody: Patient was taken into custody by the Metropolitan Police due to safeguarding issues.

Figure 5A shows the outcome of each episode of care. The most common outcome for both time periods examined was 'stay' with >70% of episodes coming into this category. Overall a downward trend in referral to hospital was seen in the postimplementation phase (figure 5B).

There were no deaths in the study.

### Health promotion activities
The change in workforce and closer relationship with colleagues and supporters enabled several health promotion activities to take place working in partnership with local services and charities. This included prostate cancer awareness, 'fit club'; a programme of activity and healthy eating, awareness of local bowel cancer screening services (as part of the national screening programme), men's health checks and offers from local smoking cessation services.[29]

### Workforce
Informal feedback is positive, and volunteer experience surveys have improved with biannual satisfaction scores improving. However, these are administered and reported centrally through the charitable body that supplies volunteers and the raw data were not available for analysis.

Although there was no formal evaluation of this (eg, satisfaction surveys), retention of the local volunteer and healthcare professional workforce is high—97% in the postimplementation phase compared with 54% in the preimplementation phase with very low turnover of staff and no attrition of the healthcare professional staff at all.

### DISCUSSION
Overall the new workforce model has met increased service demands while reducing the number of referrals to acute care. Significantly the new model uses expertise of different professional groups and involves the first aid workforce in more complex care and key decision-making, It also engaged in health promotion

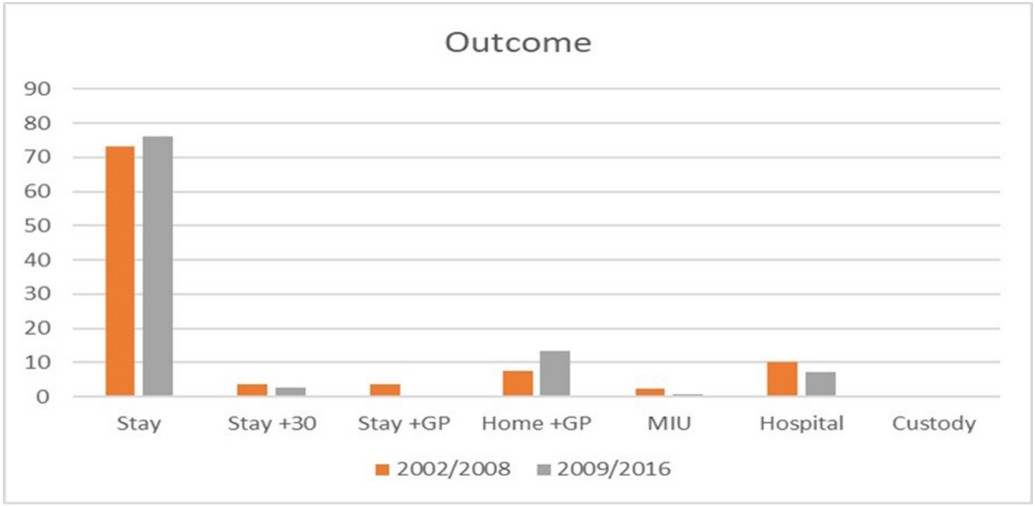

**A**

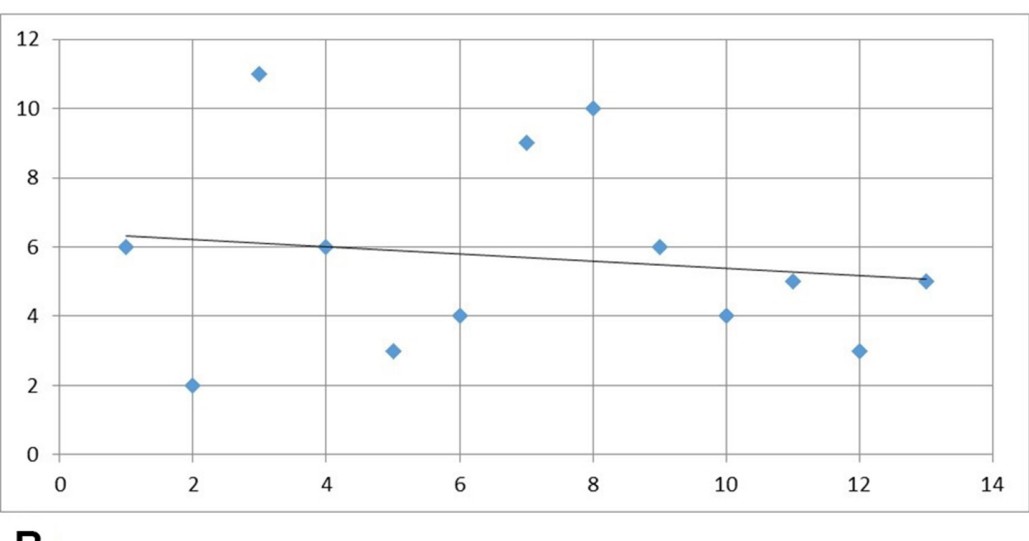

**B**

**Figure 5**  (A) Outcome of episodes of care. (B) Trend in hospital referrals by year. GP, general practitioner; MIU, Minor Injuries Unit.

activities and forging a closer working relationship with the services stakeholders.

There have been a number of incidents where the resilience of the new model has been tested, for example, with multiple concurrent casualties or serious and life-threatening incidents.[30] The response to such incidents has been swift (<3 min) with good outcomes at scene, the patients all being transferred to hospital alive. Such incidents are unusual and infrequent, requiring a combination of fundamental and advanced skills that bring together the full strength of the whole medical team.

The majority of the patients seen are of low acuity, the greater majority of presentations arise from pre-existent conditions. A significantly lesser workload arises from emergent illness or trauma, which replicates previous findings.[9–12]

In terms of leadership, the assumption that leadership comes from only one professional group[18] was never questioned. This evaluation demonstrates that wider groups of professionals other than just physicians can lead effectively and safely in these services; the service at Millwall and leadership of the medical advisory group is an example of this. It is necessary to consider the need for consistency and the attributes required in these services over and above the technical, and move to an employment model in which technical skills, experience, knowledge and leadership qualities become paramount requirements.

Only autonomous registered, regulated professionals are employed (ie, nurse practitioners who prescribe and often hold the Diploma in Immediate Care); no associate professionals are employed at the time of writing. Thus, decision-making is not an issue within the service as each group has its own code of conduct and adherence to best practice is a condition both of employment and within the 'red rules'. All professional members of the team have

appraisals and those registered with the Nursing Medical Council (NMC) or General Medical Council (GMC) are additionally subject to those regulatory bodies' revalidation requirements.

The increased workload since the introduction of the new workforce arrangements may be directly related to the new arrangements. Williams[31] posits that increased presentation rates reflect the visibility and accessibility of the medical services themselves. At Millwall the involvement of stakeholder groups and onward engagement with them, combined with joint participation in health promotion ventures at the stadium, may be influential in the increased presentation rate through increased visibility. There is evidence that crowd size in itself is not a predictor of workload.[10 11 32] Other studies show phenomenon such as an association between pitch-side performance and risk of cardiovascular events in the local population.[33]

Although this was a local evaluation, there is transferable learning to other similar environments outside of football. A flexible cohesive workforce defined by skill and driven by demand offers many advantages for all stakeholders including members of the workforce. Cohesive teams have familiarity with each other's strengths and weaknesses and can feel less coercive where 'expert power' is shared and the voices of stakeholders are heard.[34]

A flexible workforce open to other registered healthcare professionals such as nurses, doctors and paramedics with various skills and attributes allowed the team to optimise the availiability of professional skill and range of care offered. Different professional groups can perform at this level and may have attributes that are more suited to this environment. It is an important facet of the workforce model within this discussion that the registered healthcare professionals used are wider in scope than those within The Guide to Safety at Sports Grounds.[5] By including those with expertise in minor injuries and primary care, the resilience of the medical team has been optimised. This is reflected in the overall and enduring reduced referrals to external sources of care. The fact that at least one of the registered healthcare professionals at each match has experience of high-acuity prehospital care and is minimally qualified to the level of the Pre-Hospital Emergency Care Course[35] provides clinical expertise in the (rare) event of high-acuity incidents.

## CONCLUSION

The new workforce model has met increased service demands while reducing the number of referrals to acute care. It involves the first aid workforce in more complex care and key decision-making and provides a flexible registered healthcare professional team to optimise the skill mix of the team.

**Acknowledgements** The authors thank the CEO's of Millwall FC Andy Ambler and Steve Kavanagh, the directors and staff of Millwall FC, in particular Colin Sayer (stadium manager) and Jessica Newman (club secretary), Anthony Weston, Andrew Steeden and Ken Spearpoint (Millwall FC medical team), David Edwards and Siobhan Mangan (London Borough of Lewisham), the Millwall FC Safety Advisory Group and Geoff Galilee, Nikki Rutherford and Andy Robinson (Sports Ground Safety Authority) and fan advisors Melanie Bignham Attmore, Peter Garston and Micky Simpson.

**Contributors** AL contributed to the study design, interpretation, analysis and writing. AK contributed to analysis, interpretation and writing. PG, JA and JB contributed to data collection and writing. NH contributed to writing. GP contributed to study design, analysis, interpretation and writing.

**Disclaimer** The views expressed in the submitted article are those of the authors and not an official position of the employing institutions.

**Competing interests** None declared.

**Ethics approval** The local regional ethics committee deemed that ethical approval was not required for the study. This was confirmed using the HRA decision tool. In addition, changes to the service model were reviewed and approved by the local Safety Advisory Group. Permission for publication was obtained from Millwall FC and the stakeholder group.

**Provenance and peer review** Not commissioned; externally peer reviewed.

**Data sharing statement** No additional data are available.

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
