## [Reviewer comments · BMJ Open]

ARTICLE DETAILS

TITLE (PROVISIONAL)	Crowd medical services in the English Football league-remodelling the team for the 21st century using a realist approach
AUTHORS	leary, Alison; Kemp, Anthony; Greenwood, Peter; Hart, Nick; Agnew, James; Barrett, John; Punshon, Geoffrey

VERSION 1 – REVIEW

REVIEWER	Joseph Cosgrove Newcastle upon Tyne Hospitals United Kingdom Previous publications relating to Events Medicine £5000 grant from Hillsborough Family Support Group
REVIEW RETURNED	10-Aug-2017

GENERAL COMMENTS	The subject matter of this paper is becoming increasingly important in the arrangement of pre-hospital services. The authors are correct in their assertion that the original recommendations for medical cover at sports stadia came out of investigations such as the Taylor Report following the Hillsborough Disaster, April 1989. They should however also site the Bradford City FC Fire (Poplewell Inquiry) of 1985 as another driving force for such recommendations. Their assertion that the "Green Guide" does not reflect the likely reality of medical cover at football is also correct and they are to be comended for the analysis of what is likely to be expected; particularly with respect to the health of the population in the likely catchment areas for support. With respect to this latter sentence I would also urge them to consider the web-link below and the attached paper analysing cardiovascular death in 1990s Northern England in relation to football results, as means of aiding their assertion and subsequent discussion. https://www.dur.ac.uk/research/news/thoughtleadership/?itemno=24790 Another potential value of their work is that it highlights the dilemmas faced by people providing medical cover out with major stadia that host higher profile national/international events and in time it could have the potential to add to guidance for smaller sporting organisations who may find it difficult to recruit appropriate staff. The reduction in referrals to hospital is also noteworthy and should be an aim for all event medical services.
--

	In summary I would recommend reconsideration for publication in BMJ Open if a major revision of the manuscript were to occur, noting the following comments:  1. The main body of the text is too long and exceeds the 4000 words recommended in the "Guide to Authors." The "Introduction" section is very lengthy and could be effectively summarised in a briefer format. 2. The 2000 (doctor), 5000 (paramedic) guidance could be discussed in terms of potentially depriving doctors of skilled assistance when dealing with acutely ill spectators (see Smith, Cosgrove, Driscoll et al. Attached) 3. The Crowd Doctor course is now called Faculty of Pre-hospital Care Crowd Doctor Course and has recently been revised to emphasise medical management of individual spectators 4. In addition to "staff turnover" are the authors able to site any other reasons lack of team cohesiveness? (page 4, line 28) 5. For the casual reader can they clarify what they mean by high task reflexivity and lower level social reflexivity? (page 4, lines 50 & 51) 6. Are they able to provide more specifics and clarity as to why clinical leadership failed during a major incident in 2002? (Page 4, lines 54-57) And what that major incident was plus how medical staff should have been involved? 7. Throughout they mention medical professionals. Does this refer to doctors or others e.g. paramedics or both? 8. For the discussion section:  a) Decision making capacity and level or responsibility of non-doctors in terms of prescribing, referral to hospital/management within the stadia, ceasing/continuing CPR b) How have they dealt with this work in respect to appraisal and GMC/HPC revalidation?
--	--

REVIEWER	Andrew Milsten University of Massachusetts Medical School
REVIEW RETURNED	25-Aug-2017

GENERAL COMMENTS	This is a really great study, well done. It covers an important topic and showing that updates were needed and to the mass gathering regulations will be helpful for future planners. There are some revision or changes that I would suggest, that could help with the reading the paper.  1. The paper has a lot of data and is dense. I think several of the more data dense areas could be put into a table. If pressed for space, delete figures 1b and 1c. Figure 4 could be redone to show the %'s 2. Abstract: results: doesn't really say much about the comparison (PPTT rate) 3. Intro section, 3rd paragraph: #/type of providers in green guide, vs. now. Add to table 1a. Also, outside of England, people may not be familiar with the green guide 4. Intro section: 6th paragraph, interesting, but doesn't really fit in with the rest of the paper 5. Problem description section: 1st paragraph: what changes have impacted availability of crowd doctors? 6. Results, figure 2: is there a statistical difference between the PPTT pre & post implementation?
--

	It doesn't look like there is. Also, the authors mentions that the new workforce met increased service demands while reducing the numbers of referrals to acute care. So, if reading this correctly, a larger number of people came to the first-aid stations (though, not statistically significant?), but less were transferred out. Presumably, this was because of better staffing model. There is a good explanation of why first-aid stations would get more business in the discussion (5th paragraph), but the idea of increased visibility leading to more patients, is true for large outdoor festivals, not stadium events with fixed first-aid locations. 7. Did your staff treat & include players? That is not usually the case in US mass gathering literature 8. What is a "first-aider"? 9. What is a "minor injuries unit"? (urgent care?) Overall, great paper with a great design. I also like that public health interventions such as prostate screening was done. That, in of itself, would make for an interesting paper.
--	--

VERSION 1 – AUTHOR RESPONSE

Reviewer: 1

Reviewer Name: Joseph Cosgrove

Institution and Country: Newcastle upon Tyne Hospitals, United Kingdom

Please state any competing interests or state 'None declared': Previous publications relating to Events Medicine

£5000 grant from Hillsborough Family Support Group

Please leave your comments for the authors below

Comment: The subject matter of this paper is becoming increasingly important in the arrangement of pre-hospital services. The authors are correct in their assertion that the original recommendations for medical cover at sports stadia came out of investigations such as the Taylor Report following the Hillsborough Disaster, April 1989. They should however also site the Bradford City FC Fire (Poplewell Inquiry) of 1985 as another driving force for such recommendations.

Their assertion that the "Green Guide" does not reflect the likely reality of medical cover at football is also correct and they are to be comended for the analysis of what is likely to be expected; particularly with respect to the health of the population in the likely catchment areas for support. With respect to this latter sentence I would also urge them to consider the web-link below and the attached paper analysing cardiovascular death in 1990s Northern England in relation to football results, as means of aiding their assertion and subsequent discussion.

<https://www.dur.ac.uk/research/news/thoughtleadership/?itemno=24790>

Another potential value of their work is that it highlights the dilemmas faced by people providing medical cover out with major stadia that host higher profile national/international events and in time it could have the potential to add to guidance for smaller sporting organisations who may find it difficult to recruit appropriate staff. The reduction in referrals to hospital is also noteworthy and should be an aim for all event medical services.

In summary I would recommend reconsideration for publication in BMJ Open if a major revision of the manuscript were to occur, noting the following comments:

1. The main body of the text is too long and exceeds the 4000 words recommended in the "Guide to Authors." The "Introduction" section is very lengthy and could be effectively summarised in a briefer format.

RESPONSE: Thank you. This has been edited and is now slightly above 4000 words due to the additions requested by the referees. If required by the editors the word count could be reduced. The original word count was 3848.

2. The 2000 (doctor), 5000 (paramedic) guidance could be discussed in terms of potentially depriving doctors of skilled assistance when dealing with acutely ill spectators (see Smith, Cosgrove, Driscoll et al. Attached)

RESPONSE: Thank you for sending your paper. The skill mix we describe in the paper is based on demand modelling (rather than the supply side model that is promoted in the green guide). The green guide staffing recommendations appear arbitrary, non-evidence based and derived by consensus post Hillsborough. As there is no basis for the 2000 doctor or 5000 paramedic numbers it is challenging to argue for this either way. We are mindful that the role of the paramedic is not just to assist doctors. In fact we have found their skill has primacy in many situations but we are based in central London within easy reach of definitive care.

3. The Crowd Doctor course is now called Faculty of Pre-hospital Care Crowd Doctor Course and has recently been revised to emphasise medical management of individual spectators

RESPONSE: Thank you this has been amended.

4. In addition to "staff turnover" are the authors able to site any other reasons lack of team cohesiveness? (page 4, line 28)

RESPONSE: This has been clarified

5. For the casual reader can they clarify what they mean by high task reflexivity and lower level social reflexivity? (page 4, lines 50 & 51)

RESPONSE: This is based on Michael West's work. This has been clarified.

6. Are they able to provide more specifics and clarity as to why clinical leadership failed during a major incident in 2002? (Page 4, lines 54-57) And what that major incident was plus how medical staff should have been involved?

RESPONSE: This incident was widely reported in the press and is now referenced. We have made it clearer in the text that staffing was to standard in green guide including the crowd doctor role but overall staffing proved inadequate in terms of skill and attributes despite compliance with guidance. The learning was shared within the service and was a major factor in changing the workforce. It would not be appropriate to go into further detail without identifying individuals and that presents an ethical issue.

7. Throughout they mention medical professionals. Does this refer to doctors or others e.g. paramedics or both?

RESPONSE: Thank you this is doctors, nurses and paramedics primarily, we have clarified this in the manuscript.

8. For the discussion section:

a) Decision making capacity and level of responsibility of non-doctors in terms of prescribing, referral to hospital/management within the stadia, ceasing/continuing CPR

RESPONSE: Only autonomous registered, regulated professionals are employed (ie nurse practitioners who prescribe and often hold the DipIMC) we do not employ associate professionals at this time. Thus decision making is not an issue within the service as each group has its own code of conduct and adherence to best practice is a condition both of employment and within our "red rules" We have therefore addressed the leadership as part of the discussion.

b) How have they dealt with this work in respect to appraisal and GMC/HPC revalidation?

RESPONSE: Staff have regular appraisals (our staff are HCPC/NMC/GMC but only GMC/NMC require revalidation)

Reviewer: 2

Reviewer Name: Andrew Milsten

Institution and Country: University of Massachusetts Medical School

Please state any competing interests or state 'None declared': None declared

Please leave your comments for the authors below

This is a really great study, well done. It covers an important topic and showing that updates were needed and to the mass gathering regulations will be helpful for future planners. There are some revision or changes that I would suggest, that could help with the reading the paper.

RESPONSE: Thank you! We referenced your review in the original 2008 paper it was very helpful in making changes to our service.

1. The paper has a lot of data and is dense. I think several of the more data dense areas could be put into a table. If pressed for space, delete figures 1b and 1c. Figure 4 could be redone to show the %'s

RESPONSE: If requested by the editors we would be happy to do this. Adding the percentages to Figure 4 is problematic as, unless a very small font is used, the numbers overlap.

2. Abstract: results: doesn't really say much about the comparison (PPTT rate)

3. Intro section, 3rd paragraph: #/type of providers in green guide, vs. now. Add to table 1a. Also, outside of England, people may not be familiar with the green guide.

RESPONSE: Thank you. We have clarified the meaning of the Green Guide

4. Intro section: 6th paragraph, interesting, but doesn't really fit in with the rest of the paper

RESPONSE: As this is a single centre study we thought this information was important as part of the context.

5. Problem description section: 1st paragraph: what changes have impacted availability of crowd doctors?

RESPONSE: This has been addressed in the text

6. Results, figure 2: is there a statistical difference between the PPTT pre & post implementation? It doesn't look like there is.

RESPONSE: Only descriptive statistics were used.

Also, the authors mentions that the new workforce met increased service demands while reducing the numbers of referrals to acute care. So, if reading this correctly, a larger number of people came to the first-aid stations (though, not statistically significant?), but less were transferred out.

RESPONSE: That is correct .

Presumably, this was because of better staffing model. There is a good explanation of why first-aid stations would get more business in the discussion (5th paragraph), but the idea of increased visibility leading to more patients, is true for large outdoor festivals, not stadium events with fixed first-aid locations.

RESPONSE: That is our conclusion

7. Did your staff treat & include players? That is not usually the case in US mass gathering literature

RESPONSE: No we do not treat the players unless there is a very serious incident but these are rare.

8. What is a "first-aider"?

RESPONSE: Thank you we have clarified this

9. What is a "minor injuries unit"? (urgent care?)

RESPONSE: Yes it is similar to urgent care/walk in. Thank you we have clarified this

Overall, great paper with a great design. I also like that public health interventions such as prostate screening was done. That, in of itself, would make for an interesting paper.

VERSION 2 – REVIEW

REVIEWER	Dr Joseph Cosgrove Department of Perioperative and Critical Care Freeman Hospital Newcastle upon Tyne UK I have received a £5000.00 grant from the Hillsborough family Support Group to establish an Events Medicine Advisory Group
REVIEW RETURNED	07-Oct-2017

GENERAL COMMENTS	The paper's presentation has improved since the original draft however my reckoning still has the word count of the main manuscript at greater than 4000 words. Much of this could be achieved by further editing the "Introduction" section, but I would urge the authors to consider editing the word cont elsewhere in the document as well. Othe comments: Please check grammar and punctuation throughout and note use of capital letters e.g. Green guide vs. Green Guide: it would also be acceptable to italicise such terms. The Hillsborough Disaster is referred to as the "Hillsborough Disaster" and "Hillsborough Stadium Disaster" at different points in the document.
--

	"Hillsborough Disaster" should suffice as most readers will have an understanding of this. PS: my apologies for a relative delay in responding. I've had a period of ill health
--	---

REVIEWER	Andrew Milsten, MD, MS, FACEP University of Massachusetts Medical School Worcester, MA USA
REVIEW RETURNED	15-Oct-2017

GENERAL COMMENTS	Revision looks good, thank you for making the changes
---

VERSION 2 – AUTHOR RESPONSE

Reviewer One

Comment: The paper's presentation has improved since the original draft however my reckoning still has the word count of the main manuscript at greater than 4000 words. Much of this could be achieved by further editing the "Introduction" section, but I would urge the authors to consider editing the word count elsewhere in the document as well.

Other comments:

Please check grammar and punctuation throughout and note use of capital letters e.g. Green guide vs. Green Guide: it would also be acceptable to italicise such terms. The Hillsborough Disaster is referred to as the "Hillsborough Disaster" and "Hillsborough Stadium Disaster" at different points in the document. "Hillsborough Disaster" should suffice as most readers will have an understanding of this.

Response: The authors have checked the grammar and punctuation as suggested and edited the Green Guide and Hillsborough Disaster statements as requested. Some material has been removed from the paper to reduce the word count slightly .

Reviewer Two

Comment: Revision looks good, thank you for making the changes

Response: Thank you for your kind comments.